# Combined BCL-2 and PI3K/AKT Pathway Inhibition in *KMT2A*-Rearranged Acute B-Lymphoblastic Leukemia Cells

**DOI:** 10.3390/ijms24021359

**Published:** 2023-01-10

**Authors:** Clemens Holz, Sandra Lange, Anett Sekora, Gudrun Knuebel, Saskia Krohn, Hugo Murua Escobar, Christian Junghanss, Anna Richter

**Affiliations:** Department of Medicine, Clinic III—Hematology, Oncology, Palliative Medicine, Rostock University Medical Center, Ernst-Heydemann-Str. 6, 18057 Rostock, Germany

**Keywords:** acute lymphoblastic leukemia, B-ALL, BCL-2, venetoclax, apoptosis, PI3K/AKT inhibition, KMT2A, BKM-120, idelalisib, MK-2206, perifosine

## Abstract

Numerous hematologic neoplasms, including acute B-lymphoblastic leukemia (B-ALL), are characterized by overexpression of anti-apoptotic BCL-2 family proteins. Despite the high clinical efficacy of the specific BCL-2 inhibitor venetoclax in acute myeloid leukemia (AML) and chronic lymphocytic leukemia (CLL), dose limitation and resistance argue for the early exploration of rational combination strategies. Recent data indicated that BCL-2 inhibition in B-ALL with *KMT2A* rearrangements is a promising intervention option; however, combinatorial approaches have not been in focus so far. The PI3K/AKT pathway has emerged as a possible target structure due to multiple interactions with the apoptosis cascade as well as relevant dysregulation in B-ALL. Herein, we demonstrate for the first time that combined BCL-2 and PI3K/AKT inhibition has synergistic anti-proliferative effects on B-ALL cell lines. Of note, all tested combinations (venetoclax + PI3K inhibitors idelalisib or BKM-120, as well as AKT inhibitors MK-2206 or perifosine) achieved comparable anti-leukemic effects. In a detailed analysis of apoptotic processes, among the PI3K/AKT inhibitors only perifosine resulted in an increased rate of apoptotic cells. Furthermore, the combination of venetoclax and perifosine synergistically enhanced the activity of the intrinsic apoptosis pathway. Subsequent gene expression studies identified the pro-apoptotic gene *BBC3* as a possible player in synergistic action. All combinatorial approaches additionally modulated extrinsic apoptosis pathway genes. The present study provides rational combination strategies involving selective BCL-2 and PI3K/AKT inhibition in B-ALL cell lines. Furthermore, we identified a potential mechanistic background of the synergistic activity of combined venetoclax and perifosine application.

## 1. Introduction

While the overall outcome of acute lymphoblastic leukemia (ALL) has improved markedly over the past decades, the prognosis for adults (>40 years), relapsed cases, or high-risk genetic subtypes such as the *KMT2A::AFF1*-rearrangement (t(4;11)(q21;q23), *KMT2A*-r, formerly known as *MLL::AF4*), remains poor [1]. To address the insufficient response, novel therapeutical approaches based on molecular characteristics are needed. Recent studies by us and others depicted that *KMT2A*-r ALL is characterized by elevated expression of the anti-apoptotic protein BCL-2 [2,3,4]. Increased BCL-2 expression results in a disbalance favoring apoptosis avoidance and mediates chemoresistance [5,6]. 

Apoptosis can either be induced by the extrinsic pathway via extracellular receptors such as the tumor necrosis factor (TNF) receptor family or by the intrinsic mitochondrial pathway triggered by DNA damage, activated oncogenes, or nutrient deprivation [7]. As a consequence of the intrinsic pathway activation, the pro-apoptotic molecules BAD, NOXA, BIM, and BID inhibit the pro-survival proteins BCL-2, MCL-1, and BCL-xL, which ultimately disbands the inhibition of the pro-apoptotic effector molecules BAX and BAK. This BAX/BAK activation irreversibly leads to apoptosis through translocation to the outer mitochondrial membrane, subsequent release of cytochrome c, and caspase activation [6]. 

In particular, the interaction and therapeutic intervention of the intrinsic pathway has become an intensified area of oncologic research. The selective BCL-2 inhibitor venetoclax (VEN; ABT-199) demonstrated efficacy both preclinically and clinically in a broad range of hematologic neoplasms and is currently being evaluated for its clinical implication in ALL [8]. The first published phase I study to evaluate co-administration of VEN, navitoclax, and chemotherapy demonstrated encouraging results [9]. Despite notable efficacy, dose limitation as well as primary and secondary resistance to VEN monotherapy highlight the need for rational combination strategies to consistently modulate apoptotic processes in malignant cells [6,10,11]. 

Based on the physiological interaction of the BCL-2 and PI3K/AKT pathways as well as their pathologic relevance in acute leukemias, these pathways represent attractive targets for combined inhibition [12,13,14]. In addition to the direct interaction of AKT and BAD, AKT further inhibits the ubiquitin-proteasomal controlled degradation of the anti-apoptotic protein MCL-1 via phosphorylation of GSK3β and also leads to diminished p53 expression via induction of the E3 ubiquitin ligase MDM2 [13,15,16,17]. Preclinical data on simultaneous BCL-2 and PI3K inhibition in Richter’s syndrome, diffuse large B-cell lymphoma (DLBCL), and acute myeloid leukemia (AML) support the hypothesis of synergistic effects of this combinatorial approach [18,19,20]. We therefore aimed to characterize the effects of previously untested combinatorial approaches in B-ALL with *KMT2A* translocation involving the BCL-2 inhibitor VEN and the two PI3K inhibitors BKM-120 (BKM) and idelalisib (IDEL), as well as the AKT inhibitors MK-2206 (MK) and perifosine (PERI). Our data demonstrated the promising anti-leukemic potential of the tested combinations and identified possible underlying synergistic mechanisms.

## 2. Results

### 2.1. Co-Targeting BCL-2 and PI3K/AKT Pathways Synergistically Reduces Blast Viability While Sparing Healthy Blood Cells

To investigate whether novel inhibitor combinations might harm healthy blood cells, we first evaluated the potential cytotoxic effects on peripheral blood mononuclear cells (PBMC) and erythrocytes. No hemolytic activity or decrease in PBMC viability was detected (Appendix A). In contrast, the same concentrations applied on human B-ALL cell lines SEM and RS4;11 evoked decreased cell proliferation (Figure 1A). In SEM cells, all combinatorial approaches showed significantly stronger effects compared to single applications. No evident difference was observed between the combination of VEN with PI3K or AKT inhibitors. Effects on RS4;11 were not significant but comparable. Calculation of the synergism revealed that the induced effects on proliferation were synergistic in both cell lines (Figure 1B). Similar results were obtained when analyzing the metabolic activity in SEM cells. In RS4;11 cells, however, simultaneous incubation of VEN and PI3K/AKT inhibitors was not synergistic (Figure 1C,D). To identify whether synergistic effects could be modulated by the application sequence, we investigated the impact of sequential pathway inhibition. No significantly stronger regime was observed (Figure 1D). Overall, simultaneous incubation showed the most prominent results in both cell lines and was subsequently used for all following investigations. 

We further examined whether the observed effects were also reflected in the cells’ morphology (Figure 1E). Exposure of VEN caused chromatin condensation and fragmentation, which are typical characteristics of apoptosis. Within the PI3K/AKT inhibitors, PERI evoked the strongest effect in both cell lines, resulting in an increase in cell size, chromatin condensation, and vacuolization. The combinatorial approaches were in line with the results of the cell viability analysis and showed an increase in morphological changes compared to the single applications. The morphological alterations resembled characteristics of apoptotic cells. Concurrently, increased cell size and discontinuous cell membranes did not allow a clear distinction between late apoptotic or necrotic cells. Thus, further investigation of the underlying mode of action was needed.

### 2.2. Combined Application of VEN and PERI Potentiates Caspase-3 Activation and Synergistically Induces Apoptosis

Recent data supported the hypothesis that combined BCL-2 and PI3K/AKT inhibition could converge in the intrinsic apoptotic pathway [18,19,21]. We consequently characterized the rate of apoptotic cells via annexin V/PI staining (Figure 2A). As expected, the BCL-2 inhibitor VEN induced apoptosis in nanomolar concentrations in both cell lines, although not significantly in RS4;11 cells. Evaluation of the PI3K/AKT inhibitors showed that in the range of their respective IC20/30 concentrations, only PERI was able to elevate levels of annexin V+ cells. Furthermore, co-administration of VEN and PERI enhanced the apoptosis induction and revealed synergism in SEM cells (Figure 2B). Results of combined VEN and PERI incubation in RS4;11 cells were not synergistic (Figure 2B), but this was still the only approach that seemed to increase the rate of apoptotic cells compared to the single applications (Figure 2A). Combined treatment with the PI3K inhibitors IDEL or BKM induced slightly higher rates of apoptotic cells than VEN single application in SEM cells. No difference was observed after simultaneous application with the AKT inhibitor MK or with any of the inhibitors in RS4;11 cells. 

To validate that the increased rate of annexin V+ PI+ cells was triggered by apoptotic processes, we detected the level of caspase-3 cleavage. The results highlighted the special impact on apoptotic processes of PERI (Figure 2C): among the PI3K/AKT inhibitors, only PERI evoked an increased caspase-3 activation in both cell lines (not significantly in RS4;11 cells). Moreover, the application of VEN and PERI was the only combination that strongly potentiated caspase-3 cleavage in SEM cells. We observed a significant 12-fold increase in cleaved caspase-3 compared to control cells and a 3- and 1.5-fold higher caspase-3 activation than in VEN and PERI monoapplication, respectively (Appendix A). Effects on RS4;11 cells were less prominent, with only mild caspase-3 cleavage following PERI and no additive response after combined application.

This prompted us to perform a BAX translocation assay to determine whether the induction of apoptosis by PERI was dependent on the mitochondrial BCL-2 pathway activity. Co-staining of BAX and mitochondria revealed an elevated signal overlay after VEN exposure in both cell lines, which was tantamount to the induction of the mitochondrial apoptosis pathway. Using the PI3K/AKT inhibitors, PERI resulted in comparable effects while after incubation of further inhibitors, BAX and mitochondria were displayed separately. All combinatorial approaches evoked slightly elevated overlay compared to PI3K/AKT inhibitor monoapplication (Figure 2D and Appendix A). Due to severe morphological damage, the simultaneous application of VEN and PERI could not be reliably evaluated. The lack of cellular integrity resulted in molecules and organelles leaving their usual position and thus led to seemingly reduced BAX–mitochondria overlay. Given these data, a definite evaluation of whether the intrinsic apoptosis cascade was activated following VEN and PERI application was not possible. Therefore, protein and gene expression analyses needed to be conducted to elucidate the origin of the observed synergistic apoptosis induction of this and other combinations.

### 2.3. AKT Phosphorylation Correlates with Anti-Apoptotic Protein Expression and Activity

Previous studies by others and our group have shown that prolonged VEN incubation induces BCL-2, MCL-1, and BCL-xL anti-apoptotic protein expression [3,22]. Combination approaches should therefore counteract these mechanisms to synergistically tip the balance toward apoptosis induction [12]. We therefore analyzed the intracellular protein expression as well as the AKT phosphorylation to identify the potential synergistic mechanisms of combined BCL-2 and PI3K/AKT inhibition. 

No significant changes in protein expression were detected in any of the pro- or anti-apoptotic proteins analyzed (Figure 3 and Appendix A). However, it seemed that samples with reduced AKT phosphorylation also featured decreased BCL-2 phosphorylation and vice versa, especially in SEM cells. Interestingly, the PI3K inhibitor BKM induced BCL-2 phosphorylation in both cell lines, suggesting activation of the anti-apoptotic protein. Even the addition of VEN was not able to reverse this unexpected finding. Congruently, BKM and BKM + VEN samples displayed the highest pAKT and MCL-1 values of all approaches. The combined application of PERI and VEN, which demonstrated the synergistic induction of apoptosis in previous experiments (Figure 2), did not display any obvious regulation of BCL-2 pathway proteins. 

### 2.4. PERI Results in a Pro-Apoptotic Modulation of the BCL-2 Pathway That Includes Significant BBC3 Enhancement

To further study the mechanistic background of the VEN + PERI-induced synergistic apoptosis induction, we performed NGS-based custom panel RNA sequencing of 208 genes related to the PI3K/AKT and BCL-2 pathways as well as other apoptosis and leukemia signaling cascades. The principal component analysis as well as total read counts revealed that the SEM and RS4;11 cells were characterized by a differential basal gene expression pattern (Appendix A). 

Hierarchical clustering of all investigated genes demonstrated that VEN incubation was structured along with the controls in both cell lines (Appendix A). Overall, no significant change compared to single application was observed following combined inhibition. Remarkably, approaches with PERI clustered apart from other PI3K/AKT inhibitors in SEM cells. In contrast, RS4;11 cells had a more homologous expression structure without distinct PERI clustering.

A further focus on the PI3K/AKT and BCL-2 pathways identified that the upregulation of *MDM2* in RS4;11 cells was the strongest modulation within the PI3K/AKT pathway (Figure 4A). Interestingly, *MDM2* was induced by all PI3K/AKT inhibitors except for PERI, again suggesting a distinct mode of action of this inhibitor. No comparable induction of *MDM2* gene expression was observed in SEM cells or after VEN incubation. Generally, no consistent modulation of the PI3K/AKT pathway in terms of the biological function was identified in any approach in both cell lines. In addition, PI3K/AKT genes were frequently regulated in adverse ways in SEM and RS4;11 cells, suggesting different modes of action of the inhibitors in both cell lines.

Regarding the gene expression modulation of the BCL-2 pathway, only PERI caused a predominantly pro-apoptotic effect, while other PI3K/AKT inhibitors evoked ambiguous effects (Figure 4B). Only incubation with PERI slightly downregulated the anti-apoptotic BCL-2 family genes *BCL2*, *MCL1*, and *BCL2L1* (BCL-xL). The most outstanding was the PERI-mediated induction of the pro-apoptotic BH3-only gene *BBC3* in SEM cells, translating into the PUMA protein. Within the filtered 176 genes with an average total read count >50 to justify biological significance, the upregulation of *BBC3* after PERI exposure (fold change of 3.82) was the strongest modulation of all genes and approaches measured in our study (Figure 4C). To identify a possible explanation of why the prominent *BBC3* regulation was restricted to SEM cells, we analyzed the basal expression of the *BBC3* upstream mediator p53. Matching the picture, RS4;11 cells did not express any p53 protein while SEM cells showed strong p53 signals (Appendix A; for the original full-length immunoblot, see Appendix A).

### 2.5. PI3K/AKT Inhibitors Cause Modulation of the Extrinsic Apoptosis Signaling Cascade

Our findings demonstrated that PERI has an exceptional role within the PI3K/AKT inhibitors in the pro-apoptotic modulation of the intrinsic pathway. In the next step, we sought to determine further effects on the wider apoptotic signaling cascade (Figure 5). Here, it became evident that the evoked effects were not restricted to the mitochondrial pathway, especially in SEM cells. PERI induced activators (*TNF* and *FAS*), adaptors (*FADD* and *TRADD*), and an effector (*CASP10*) of the extrinsic apoptotic signaling. The strongest induction within the extrinsic signaling pathway was observed for the *TNF* gene, but also for other members of the TNF superfamily; e.g., *TNFSF10* or *TNFRSF1A* showed increased expression. In comparison, VEN application resulted in modest effects on the gene expression of the extrinsic apoptotic signaling cascades (Appendix A), and no relevant differences were subsequently observed after the combined application of VEN and PERI (Appendix A). The results observed in SEM cells were not specific to the application of PERI. All other PI3K/AKT inhibitors also resulted in an induction of extrinsic signaling genes, including the increased expression of activators, adaptors, and effectors (Appendix A). 

In contrast, no evident pro-apoptotic modulation was detected in the RS4;11 cell line. Even though PERI showed the strongest tendency within all approaches to activate the extrinsic apoptotic pathway, the results were ambiguous. While gene expressions of most members of the TNF superfamily were increased following PERI incubation, the activator (*FAS*), adaptors (*FADD* and *TRADD*), and the gene expression of *CASP8* and *CASP10* were diminished. 

## 3. Discussion

At an early stage of clinical assessment of VEN in ALL patients, we evaluated rational combination strategies. In this study, we investigated the effects of combined BCL-2 and PI3K/AKT inhibition for the first time in human B-ALL cell lines.

Matching the data of clinical studies demonstrating manageable safety profiles of all inhibitors used in our study [23,24,25,26,27], initial analyses showed no increased risk of cytotoxic effects on healthy blood cells in vitro. Our results depicted that targeted BCL-2 and PI3K/AKT inhibition acted synergistically in two human B-ALL cell lines with *KMT2A::AFF1* translocation, which matched the findings of other groups who investigated the efficacy of simultaneous BCL-2 and PI3K inhibition in DLBCL, Richter’s syndrome, non-Hodgkin lymphoma (NHL), and AML cells [18,19,20,28]. Still, due to the limited number of cell lines included in the study, further investigation will be indispensable for a general statement regarding the practicability of those combinations in *KMT2A*-r B-ALL. The observed anti-leukemic effects were independent of the selectivity or localization of PI3K/AKT pathway inhibition. 

By examining apoptotic processes, we revealed the exceptional role of PERI, the only PI3K/AKT inhibitor that induced apoptosis as single agent using IC20/30 concentrations in SEM and RS4;11 cells. Other work indicated that BKM and IDEL induced apoptosis at higher concentrations in ALL cell lines [29,30]. In line with our findings, a study on human AML cell lines demonstrated that incubation with MK evoked anti-proliferative effects in low concentration ranges (0.1–0.2 µM), while 5–10 µM were necessary to achieve an increased rate of apoptotic cells [31]. In the context of all results of the present study, co-administration of VEN and PERI was the only approach that induced a significant and synergistic increase in apoptosis. This finding was restricted to SEM cells and raised questions about different mechanisms of the inhibitors and differential effects between cell lines. 

PERI is an orally available phospholipid analog. Despite specific AKT inhibition, interactions with cellular membranes, lipid rafts, and phospholipid metabolism occur and might cause stronger effects [32]. A phase II clinical trial in relapsed/refractory CLL that included in vitro studies demonstrated AKT-independent cytotoxicity of PERI [23]. In line with those results, we observed variable pAKT expression after PERI incubation in SEM cells, thereby supporting the hypothesis of AKT-independent mechanisms. Additionally, we have already shown that MK exhibits effects independently of the AKT phosphorylation status [33]. The selectivity of AKT inhibitors remains a great challenge [34]. BKM also seems to induce AKT-independent mechanisms as well as activation of compensatory signaling pathways as demonstrated by increased AKT phosphorylation and pBCL-2 expression. Similarly, a phase I study that evaluated the efficacy of BKM in acute leukemias revealed increased pAKT expression in the presence of decreased phosphorylation of downstream substrates [35]. 

In agreement with previous studies of PERI in multiple myeloma, AML, and T-ALL cell lines, we verified modulated activity of the intrinsic and extrinsic apoptosis pathways [36,37,38]. Synergistic effects of the combination of VEN and PERI may converge at multiple points of apoptosis. Except for the co-administration of PERI with the dual BCL-2 and BCL-xL inhibitor navitoclax in lung cancer cell lines, there are hardly any data on combinatorial approaches. In this study, synergistic activity of PERI and navitoclax was linked to PERI-mediated degradation of MCL-1 [39]. Interestingly, all studies that investigated combined BCL-2 and PI3K inhibition related a major part of the synergistic activity to the diminished expression levels of the anti-apoptotic proteins MCL-1 or BCL-xL irrespective of the hematological neoplasm (DLBCL, Richter’s syndrome, NHL, and AML) [18,19,20,28]. Still, we could not prove this interaction in two B-ALL cell lines on the gene or protein expression level. 

In line with the apoptosis experiments, the gene expression analysis further showed that PERI stood out from other inhibitors. PERI induced a pro-apoptotic modulation of the intrinsic apoptotic signaling cascade, which matched existing data in other hematological entities [36,37,38]. PERI uniquely regulated the pro-apoptotic gene *BBC3* (PUMA) in SEM cells. The BH3-only protein PUMA is an unselective inhibitor of anti-apoptotic BCL-2 proteins and is induced by the tumor suppressor p53 [6,40,41]. We observed this exceptional regulation only in SEM cells, which harbored strong basal p53 expression, while no regulation of *BBC3* was seen in p53-deficient RS4;11 cells. The different expression of p53 could provide one possible explanation for the different behavior of both cell lines. Interestingly, while *BBC3* was significantly upregulated in SEM cells, no relevant modulation of *TP53* was observed on the gene expression level. This discrepancy could possibly be explained by p53-independent regulation of *BBC3*: Recent studies reported that members of the JAK/STAT (STAT3) [42] and PI3K signaling (AKT, FOXO1, FOXO3A, and GSK3β) [43,44,45,46,47,48] pathways as well as Hippo (YAP1) [49] and MAPK cascade members (ERK, p38, and c-JUN) [50,51,52] contributed to the modulation of *BBC3* expression. However, none of these molecules exhibited relevant changes in gene expression following PERI incubation. However, there are possible p53-independent regulators of *BBC3* that were not included in our gene expression panel; for example, members of the TGF-β signaling pathway [53,54]. Those players, along with potential regulation at the epigenetic or post-translational level or modulation of feedback loops, might be responsible for the observed simultaneous *BBC3* upregulation without an influence on *TP53*. It is also possible that *BBC3* is a direct target of PERI. Although the upregulation of *BBC3* was the strongest modulation observed in the entire RNAseq panel and thus suggests a high biological relevance, the current data were restricted to gene expression and must be validated in future studies.

In conclusion, we investigated for the first time the effects of combined BCL-2 and PI3K/AKT inhibition in human *KMT2A*-r B-ALL cell lines, thereby proving synergistic anti-leukemic activity. Analyses of protein and gene expression revealed previously undescribed mechanisms for synergistic efficacy of the selective BCL-2 inhibitor VEN and PI3K/AKT inhibition. The combination of VEN and PERI differed from other approaches in terms of the synergistic induction of apoptotic processes. However, further studies are needed to validate these effects in vivo. The clinical application of PERI in Waldenstrom macroglobulinemia and CLL showed limited efficacy with mainly stable disease but was overall well tolerated [23,55]. When used in combination, feedback mechanisms, resistance, and dose limitation could be minimized, thereby overcoming limitations. 

## 4. Materials and Methods

### 4.1. Cell Lines and Inhibitors

The human B-ALL cell lines SEM and RS4;11 were obtained from DSMZ (Braunschweig, Germany) and cultivated as previously described [3]. A control for authenticity (cell surface flow cytometry) and mycoplasma contamination was performed at regular intervals. Venetoclax (VEN), idelalisib (IDEL), and perifosine (PERI) were purchased from Hycultec (Beutelsbach, Germany). BKM-120 (BKM) and MK-2206 (MK) were obtained from Selleckchem (Houston, TX, USA). All inhibitors were dissolved in DMSO and stored at −80 °C.

### 4.2. Inhibitory Experiments

Study groups were not blinded to the investigators. Inhibitors were applied individually or in combination for 24–72 h in at least three biological replicates. Incubation with DMSO served as control to exclude effects being evoked by the solvent. Cell proliferation was analyzed via trypan blue staining after 72 h incubation. The metabolic activity was assessed via WST-1 assay (Roche, Mannheim, Germany) in technical triplicates after 72 h incubation. Effects of sequential application regimes were determined when VEN was added 24 h before the PI3K/AKT inhibitor or inversely, and data were analyzed 72 h after the first reagent was added. Approaches were performed with concentrations that resulted in a proliferation decrease of ca. 20–30% (IC 20/30 concentrations) in both cell lines (SEM: VEN, 10 nM; BKM, 750 nM; IDEL, 10 µM; MK, 250 nM; PERI, 5 µM/RS4;11: VEN, 2.5 nM; BKM, 500 nM; IDEL, 10 µM; MK, 250 nM; PERI, 7.5 µM). 

To ensure that previously untested combinations had no toxicity in non-neoplastic blood cells, effects on viable blood cells of five healthy voluntary donors were evaluated using a Calcein AM and hemolysis assay as previously described [33]. Inhibitors were applied individually as well as simultaneously using the highest concentrations used in the cell culture experiments. Results of proliferation and cytotoxicity assessment were compared to our own previously published data on combined MK and VEN incubation [33]. 

### 4.3. Morphology Analyses

SEM and RS4;11 cells were incubated with inhibitors for 48 h and processed as previously described [3]. 

### 4.4. Analysis of Apoptotic Processes

For flow-cytometric apoptosis assessment, cells were incubated with VEN and PI3K/AKT inhibitors for 72 h, treated as previously described [3], and analyzed using the FACSCalibur device (Becton Dickinson, Heidelberg, Germany). Data were evaluated using FACSuite software (Becton Dickinson, version 1.0.6.5230). Here, annexin V+ PI− cells were classified as early apoptotic and annexin V+ PI+ cells as late apoptotic or necrotic. Results were compared to our own previously published data on combined MK and VEN incubation [33]. Translocation of the effector protein BAX to the outer mitochondrial membrane was investigated via an immunofluorescence-based BAX translocation assay as previously described [33]. 

### 4.5. Intracellular Flow-Cytometric Protein Expression Measurement

Cleaved caspase-3 and BCL-2 family protein expression as well as AKT phosphorylation was measured via intracellular staining with antibody-coupled fluorophores and subsequent detection using flow cytometry as previously described [3]. The following antibodies were used according to the manufacturer’s guidelines: BCL-2-PerCP-Cy5.5 clone C-2 (cat. sc-7382); p-BCL-2-Alexa Fluor-647 clone A-11 (cat. sc-377554); MCL-1-PE clone 22 (cat. sc-12756); BCL-XL-PerCP-Cy5.5 clone H-5 (cat. sc-8392); Bax-PE clone 2D2 (cat. sc-20067; all Santa Cruz Biotechnology (Dallas, TX, USA)), Cleaved Caspase-3-Alexa Fluor-488 polyclonal (cat. 9669; Cell Signaling, Danvers, MA, USA); and pAKT-Alexa Fluor^®^ 488 conjugate (cat. 4071; Cell Signaling).

### 4.6. Targeted RNA Custom Panel Sequencing

Targeted gene expression analysis was performed with a custom designed panel of relevant BCL-2, PI3K/AKT, and apoptotic pathway genes as previously described [3]. Expression values and fold change (FC) were calculated using the Transcriptome Analysis Console software (TAC, version 4.0.2, Thermo Fisher Scientific, Waltham, MA, USA). To optimally represent the FC in linear space, FC values between 0 and 1 were modified by the TAC software as follows: (−1/Fold change)).

### 4.7. Immunoblot

Analysis of basal p53 expression was performed via immunoblot as previously described [33] using the primary antibody p53 (1C12) mouse mAb IgG (cat. 2524; Cell Signaling, 1:1000 dilution) and secondary antibody IRDye^®^ 680RD goat anti-mouse IgG (cat. 926-68070; LI-COR Biosciences, Lincoln, NE, USA; 1:5000 dilution). Blots were processed and cropped using Image Studio Lite 5.2 software and MS PowerPoint (2011) to improve clarity and conciseness.

### 4.8. Statistical Analyses

To generate valid and reproducible results, all experiments were performed at least in biological and technical triplicates if not otherwise indicated. All values are expressed as mean ± standard deviation. For statistical analysis, Gaussian normality distribution was tested in all cases using a Shapiro–Wilk test to determine the following statistical procedure. Statistical testing was performed using GraphPad PRISM software (version 8); the exact test used is indicated in the respective figure legends. Statistical significance was defined as * *p* < 0.05, ** *p* < 0.005, and *** *p* < 0.001. 

Synergistic effects were calculated according to the Bliss independence model [56]. Results of the combined treatment were compared to an expected effect (E). The calculation of E was based on the inhibition of the single agents A and B as follows: E = (A + B) − (A × B). The difference between the observed (O) and expected effect (∆ = O − E) defined the degree of synergy of the combined treatment. Values higher than zero (∆ > 0) indicated synergistic effects, while a difference ∆ < 0 implied antagonism. The degree of synergy/antagonism is depicted as color-coded tiles with green and red standing for synergy and antagonism, respectively. Across all figures (including Bliss values), the same color scale was used to allow for easy comparison of the synergistic effects throughout different analyses. The top and bottom ends of the scale were tantamount to the highest calculated value and therefore were set to 0.281 and −0.281, respectively.

## Figures and Tables

**Figure 1 ijms-24-01359-f001:**
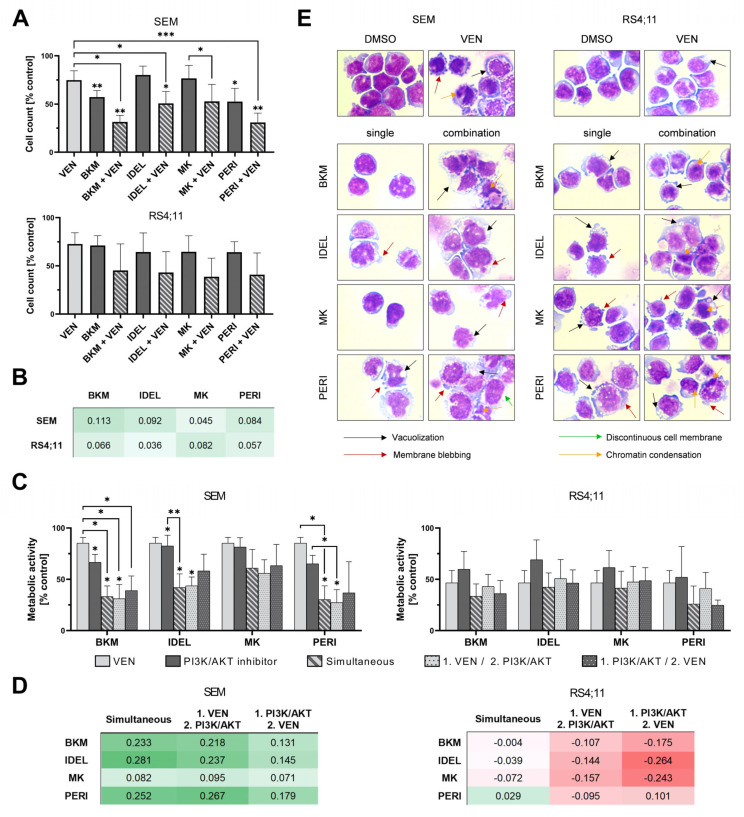
Effects of combined BCL-2 and PI3K/AKT inhibition on cell viability and morphology. (**A**) Proliferation was examined via trypan blue staining after 72 h incubation (n ≥ 3; ANOVA and post-hoc Tukey’s multiple comparison test, * *p* < 0.05, ** *p* < 0.005, *** *p* < 0.001). (**B**) Analysis of synergistic anti-proliferative approaches using the Bliss independence model for data displayed in (**A**). Values ∆ > 0 (green) indicate synergism while ∆ < 0 (red) implies antagonistic effects. (**C**) The metabolic activity was analyzed via WST-1 assay after 72 h incubation. For sequential regimens, VEN was added 24 h before the PI3K/AKT inhibitor (1. VEN/2. PI3K/AKT) or inversely (1. PI3K/AKT/2. VEN) (n ≥ 3; ANOVA and post-hoc Tukey’s multiple comparison test, * *p* < 0.05, ** *p* < 0.005, *** *p* < 0.001). (**D**) Evaluation of synergistic effects on metabolic activity using the Bliss model for data presented in (**C**). Values ∆ > 0 (green) indicate synergism while ∆ < 0 (red) implies antagonistic effects. (**E**) Assessment of cell morphology after 48 h incubation using Pappenheim staining. Representative images of three independent biological replicates at 100× magnification.

**Figure 2 ijms-24-01359-f002:**
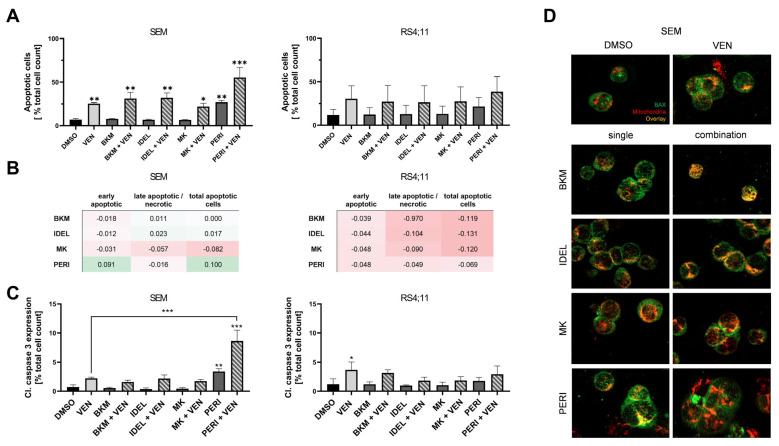
Analysis of apoptotic processes following combined BCL-2 and PI3K/AKT inhibition. (**A**) Flow-cytometric annexin V/PI analysis after 72 h incubation (n ≥ 3; ANOVA and post-hoc Tukey’s multiple comparison test, * *p* < 0.05, ** *p* < 0.005, *** *p* < 0.001). (**B**) Analysis of synergistic apoptosis induction via Bliss independence model using the data presented in (**A**). Values ∆ > 0 resemble synergism (green) while ∆ < 0 implies antagonistic effects (red). (**C**) Detection of activated (cleaved) caspase-3 via intracellular flow cytometry after 48 h incubation (n ≥ 3; ANOVA and post-hoc Tukey’s multiple comparison test, * *p* < 0.05, ** *p* < 0.005, *** *p* < 0.001). (**D**) Immunofluorescence-based BAX translocation assay following 48 h inhibitor incubation. Representative images of at least three independent biological and technical replicates at 400× magnification.

**Figure 3 ijms-24-01359-f003:**
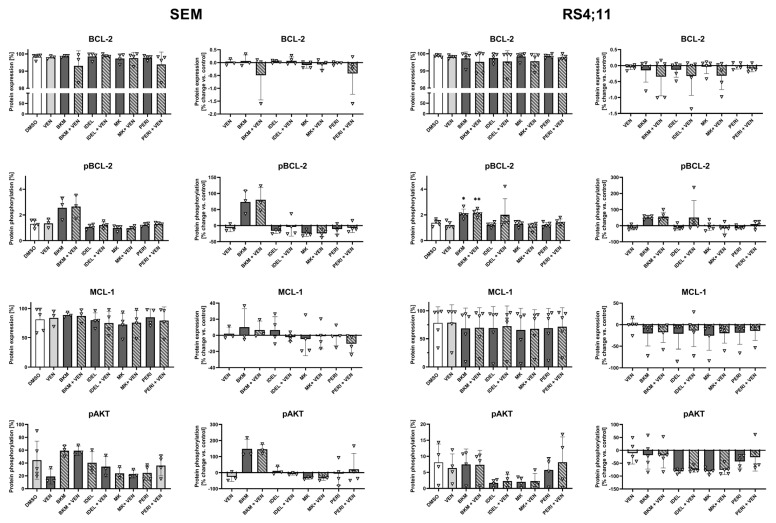
Detection of protein expression and activity using intracellular flow cytometry. Cells were incubated with VEN and PI3K/AKT inhibitors for 48 h. Representation of the absolute protein expression/phosphorylation (left graphs) indicating the amount of target-positive cells within the entire cell population. The relative proportion of positive cells compared to control cells was calculated within each replicate and is represented in the right graphs (n ≥ 3; ANOVA and post-hoc Tukey’s multiple comparison test, * *p* < 0.05, ** *p* < 0.005).

**Figure 4 ijms-24-01359-f004:**
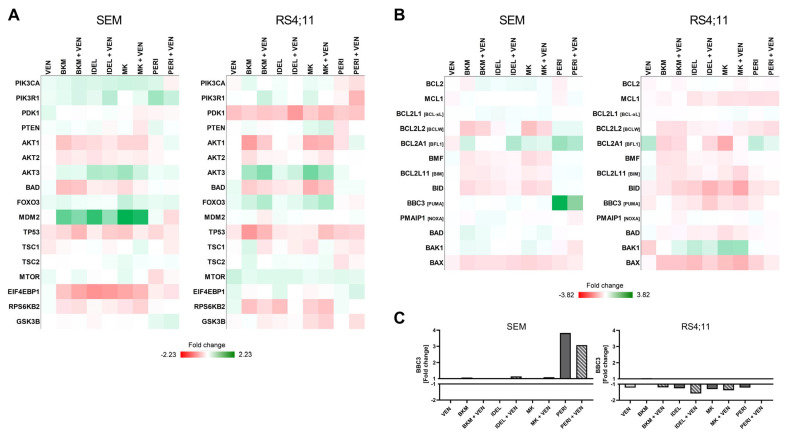
Analysis of PI3K/AKT and BCL-2 pathway gene expression via targeted RNA panel sequencing. (**A**,**B**) Heatmap of the PI3K/AKT (**A**) and BCL-2 (**B**) pathway gene expression. Fold change (FC) compared to control. Green and red colors represent up- and downregulation, respectively. (**C**) Bar graph of *BBC3* fold change after treatment compared to control. FC values may be covered by the scale of the graph and are therefore additionally listed in Appendix A.

**Figure 5 ijms-24-01359-f005:**
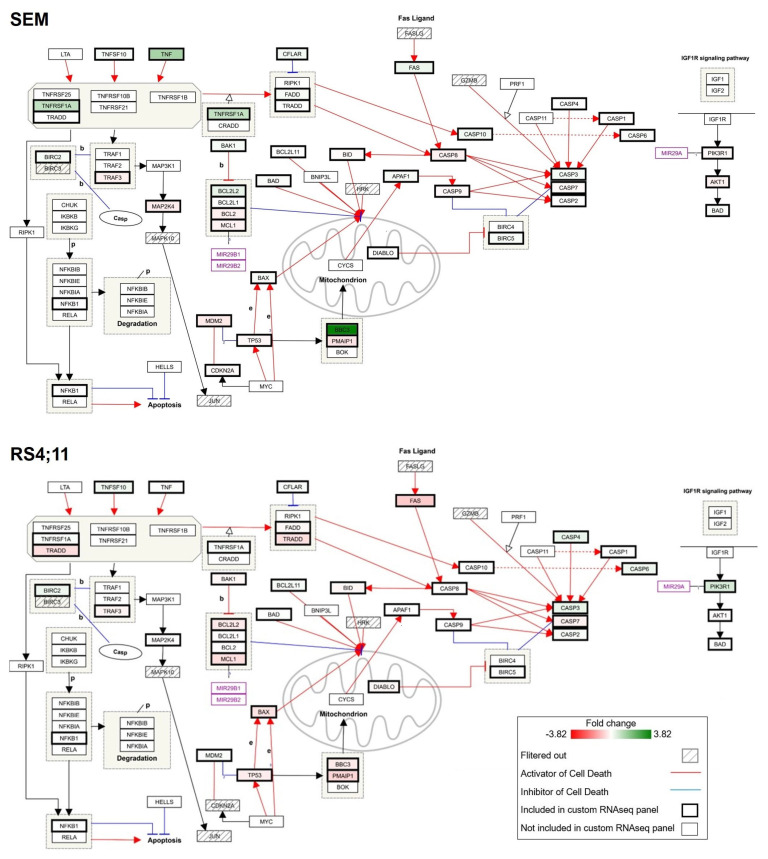
Effects of PERI administration on gene expression of the apoptosis signaling cascade. After 24 h of inhibitor incubation, gene expression was examined via targeted RNA sequencing. Subsequent data analysis was performed using TAC software. The WikiPathways plugin was used to integrate the FC expression values within the apoptosis signaling cascade. Green and red colors represent up- and downregulation, respectively. Genes with an average total read count < 50 were excluded to justify biological significance (labeled as filtered out).

## Data Availability

The datasets used and/or analyzed during the current study are available from the corresponding author upon reasonable request.

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
