# Peer review of "Combined BCL-2 and PI3K/AKT Pathway Inhibition in KMT2A-Rearranged Acute B-Lymphoblastic Leukemia Cells"

_ijms, 2023, doi:10.3390/ijms24021359_

Round 1

Reviewer 1 Report

In this study co-treatment of B-ALL cell lines with BCL-2 and PI3K/AKT causes synergistic antiproliferative effects. They utilize a number of assays to validate their hypothesis.  In addition, the authors used gene expression studies and  identified the pro-apoptotic gene BBC3 as a possible player of  synergistic action. The study concludes that a potential mechanistic background of the synergistic activity of combined venetoclax and perifosine application. This is a well-planned and executed  investigation. However, there are the following issues to be clarified before the publication

1.    How the drugs were dissolved?

2.    Why is DMSO used as control?

3.    Please explain IC 20/30 concentrations

4.    The data of co-treatment with BCL-2 and PI3K/AKT inhibitor on non-neoplastic cells should be shown in Figure 2, not in Supplementary material

5.    In most of the bar graphs p-value is mission. For example, in Figures 1 A and C, the information on the p-value is missing. A similar case in Figure 2 and others

6.    Presentation of supplementary Figure S5 is poorly presented. The heat map is not showing potential upregulated and downregulated genes. It needs to be corrected and presented with genes

7.    The supplementary Figure 7, the western blot is not labeled properly with the treatment condition and molecular weight of the protein

Author Response

Response letter to Reviewer 1

We thank reviewer 1 for the careful revision of our manuscript as well as the many suggestions for improvement. We have now revised the paper and think that it greatly benefits from the corrections made by reviewer 1. All changes are documented in track changes in the revised version of the paper. We hope that the manuscript is now suitable for publication in IJMS.

Reviewer 1: How the drugs were dissolved?

Answer: We apologize for this missing information and added the respective sentence in the Materials and Methods section: “All inhibitors were dissolved in DMSO and stored at ‑80°C.” (line 324-325).

Reviewer 1: Why is DMSO used as control?

Answer: We used DMSO as control for the inhibitor studies because all inhibitors were dissolved in DMSO. Therefore, we are able to exclude that the effects we observed were evoked by the solvent instead of the inhibitor itself. We have now clarified this topic in the Materials and Methods section: “Incubation with DMSO served as control to exclude effects being evoked by the solvent.“ (line 328-329).

Reviewer 1: Please explain IC 20/30 concentrations

Answer: The term “IC20/30” is a commonly used term to describe the concentration of an inhibitor that results in cell death in approximately 20 to 30 % of the cells. We have included this explanation in the Materials and Methods section: “Approaches were performed with concentrations resulting in a proliferation decrease of ca. 20-30 % (IC 20/30 concentrations) in both cell lines“ (line 334-335).

Reviewer 1: The data of co-treatment with BCL-2 and PI3K/AKT inhibitor on non-neoplastic cells should be shown in Figure 2, not in Supplementary material

Answer: Regarding this topic, we disagree with Reviewer 1. The figures in the main paper should only contain the most important data that is necessary to understand the main findings of the research. In our case, this would be the inhibitor experiments elucidating the effects of combined BCL-2 and PI3K/AKT inhibition in leukemia cells. The results of the inhibitors’ effects on healthy cells, on the other hand, are important to rule out potential side-effects. All tested drugs were, however, previously assessed in clinical studies and effects on healthy PBMCs and erythrocytes were therefore not not expected. Due to these reasons, we would prefer to keep those data in the supplement. Should the Reviewer or Editor insist on the figure being moved to the main part of the manuscript, we would of course follow this suggestion.

Reviewer 1: In most of the bar graphs p-value is mission. For example, in Figures 1 A and C, the information on the p-value is missing. A similar case in Figure 2 and others 

Answer: In the Materials and Methods section, part Statistical analyses, the meaning of the asterisks is stated as follows: “Statistical significance was defined as * P<0.05; ** P<0.005 and *** P<0.001.” (line 388-389) Again, should the Reviewer or Editor wish to add this information in the respective figure legends, this is of course possible.

Reviewer 1: Presentation of supplementary Figure S5 is poorly presented. The heat map is not showing potential upregulated and downregulated genes. It needs to be corrected and presented with genes

Answer: The current way of data presentation represents a common way and is used in this form for years. Due to the high number of genes analyzed, it is not practicable to enter every gene name at the side. Also, the aim of this figure was to demonstrate that controls (blue) and venetoclax-treated samples (red) showed a similar gene expression pattern compared to the PI3K/AKT inhibitors and that perifosine-treated samples clustered apart from all other approaches. This is obvious without knowing the exact gene names in the heatmap.

Reviewer 1: The supplementary Figure 7, the western blot is not labeled properly with the treatment condition and molecular weight of the protein

Answer: We apologize for the missing information and have now properly labelled the size standard and protein sizes.

Author Response

Response letter to Reviewer 2

We thank reviewer 2 for the revision of our manuscript as well as the suggestions for improvement. We have now revised the paper according to the comments of all reviewers and think that it clearly benefits from the corrections made. All changes are documented in track changes in the revised version of the paper. We hope that the manuscript is now suitable for publication in IJMS.

Reviewer 2: The authors have not discussed the role of BIM in B-ALL cell lines having high BCL-2 expression with regards to venetoclax treatment. Venetoclax targets BCL2 to disrupt BIM:BCL-2 complex and induces BIM-mediated apoptosis in SCLC cell lines. Does it do similar in B-ALL cell lines. It can be shown by pulling down BIM and probing to BCL-2 to see if there is disruption in BIM:BCL-2 complex by venetoclax treatment. If BIM plays a role in apoptosis with to regards to Venetoclax treatment it also better to do additional rescue experiment with siBIM.

Answer: We thank reviewer 2 for bringing the importance of BIM for venetoclax efficacy in SCLC to our attention. There are a couple of studies in B-ALL that investigated the role of BIM in venetoclax-mediated apoptosis induction. However, those studies focused on either BCR::ABL1 positive ALLs (Scherr et al., Leukemia 2019) or hypodiploid ALL cell lines (Diaz-Flores et al., Cancer Research 2019). The first study found that in BCR::ABL1 positive ALLs, the BIM expression is generally low. As also correctly suggested by reviewer 2, venetoclax exposure resulted in BIM displacement from BCL2 in responsive cell lines and thus induced apoptosis. Accordingly, BIM knockdown lead to reduced venetoclax cytotoxicity. The second study also found that samples and cell lines with high BCL2 and BIM expression were responsive to venetoclax. Interestingly, they also discovered that some cells lacking BIM also responded well to venetoclax, suggesting an alternative mode of action. They discovered upregulation of cell arrest as well as induction of apoptosis mediators as possible mechanisms.

In our study, we used the KMT2A::AFF1 rearranged cell lines SEM and RS4;11 for our analyses. As shown in Figure S4, both cell lines show rather high BIM (BCL2L11) gene expression levels. It is therefore very likely that BIM plays a major role for venetoclax-mediated apoptosis induction. Further, no relevant up- or down-regulation of BIM was observed following either venetoclax or combined application in our experiments. The combined inhibition of BCL2 (by venetoclax) and PI3K/AKT pathway members was the actual scope of our work. We tried to identify possible modes of synergism and found the upregulation of BBC3 as a likely mechanism. Due to the high amount of data generated, the limited space available, and the lack of relevance of BIM for our research question, we did not focus on BIM in our manuscript. The experiments suggested by reviewer 2 (BIM protein pulldown and siRNA) are very interesting options to further elucidate the role of BIM in both, venetoclax as well as PI3K/AKT inhibitor efficacy in KMT2A rearranged B-ALL cell lines. They are, however, behind the scope of the present manuscript.

Round 2

Reviewer 1 Report

"however, previously assessed in clinical studies and effects on healthy PBMCs and erythrocytes were therefore not expected". 

Can author cite a reference for this statement in the manuscript

Author Response

Response letter to Reviewer 1

We again thank reviewer 1 for the careful revision of our manuscript. We have now revised the paper. We hope that the manuscript is now suitable for publication in IJMS.

Reviewer 1: "however, previously assessed in clinical studies and effects on healthy PBMCs and erythrocytes were therefore not expected". 

Can author cite a reference for this statement in the manuscript

Answer: This is a valid point made by reviewer 1 and we fully agree that this should be mentioned in the manuscript. We have therefore included a respective sentence and of course matching references in the discussion section of the paper: “Matching the data of clinical studies demonstrating manageable safety profiles of all inhibitiors used in our study [23-27], initial analyses showed no increased risk of cytotoxic effects on healthy blood cells in vitro.” (line 228-230).